# Identification of Apple Flower Development-Related Gene Families and Analysis of Transcriptional Regulation

**DOI:** 10.3390/ijms25147510

**Published:** 2024-07-09

**Authors:** Chuang Mei, Xianguo Li, Peng Yan, Beibei Feng, Aisajan Mamat, Jixun Wang, Ning Li

**Affiliations:** 1The State Key Laboratory of Genetic Improvement and Germplasm Innovation of Crop Resistance in Arid Desert Regions (Preparation), Key Laboratory of Genome Research and Genetic Improvement of Xinjiang Characteristic Fruits and Vegetables, Institute of Horticultural Crops, Xinjiang Academy of Agricultural Sciences, Urumqi 830091, China; meichuangxj@163.com (C.M.); 15099198840@163.com (X.L.); xaasyysyp@163.com (P.Y.); fengbeibei@xaas.ac.cn (B.F.); aisajan_116@163.com (A.M.); 2The State Key Laboratory of Genetic Improvement and Germplasm Innovation of Crop Resistance in Arid Desert Regions (Preparation), Urumqi 830091, China

**Keywords:** C2H2, DELLA, FKF1, evolution, floral development

## Abstract

Apple (*Malus domestica* Borkh.) stands out as a globally significant fruit tree with considerable economic importance. Nonetheless, the orchard production of ‘Fuji’ apples faces significant challenges, including delayed flowering in young trees and inconsistent annual yields in mature trees, ultimately resulting in suboptimal fruit yield due to insufficient flower bud formation. Flower development represents a pivotal process influencing plant adaptation to environmental conditions and is a crucial determinant of successful plant reproduction. The three gene or transcription factor (TF) families, C2H2, DELLA, and FKF1, have emerged as key regulators in plant flowering regulation; however, understanding their roles during apple flowering remains limited. Consequently, this study identified 24 *MdC2H2*, 6 *MdDELLA*, and 6 *MdFKF1* genes in the apple genome with high confidence. Through phylogenetic analyses, the genes within each family were categorized into three distinct subgroups, with all facets of protein physicochemical properties and conserved motifs contingent upon subgroup classification. Repetitive events between these three gene families within the apple genome were elucidated via collinearity analysis. qRT-PCR analysis was conducted and revealed significant expression differences among *MdC2H2-18*, *MdDELLA1*, and *MdFKF1-4* during apple bud development. Furthermore, yeast two-hybrid analysis unveiled an interaction between *MdC2H2-18* and *MdDELLA1*. The genome-wide identification of the C2H2, DELLA, and FKF1 gene families in apples has shed light on the molecular mechanisms underlying apple flower bud development.

## 1. Introduction

Flowering, a pivotal phase dictating the reproductive success of angiosperms, necessitates the precise integration of internal and environmental cues to initiate the process. Flower development profoundly influences plant adaptation and species perpetuation, particularly in the realm of plant production, understanding the characteristics and dynamics of crop flower development is virtually the cornerstone of all agricultural activities. Insights gleaned from the study of angiosperms have revealed a multifaceted regulatory network governing the flowering process. Six primary regulatory pathways have been delineated, encompassing photoperiodism [1], autonomy [2], vernalization [3], gibberellic acid (GA) [4], temperature, and age signaling [5,6].

The Fuji apple (*Malus domestica* Borkh. cv. Red Fuji) represents a significant cultivar in China’s apple industry. However, challenges such as delayed flowering in young trees and inconsistent annual yields in mature trees are prevalent. Understanding the genetic underpinnings and regulatory mechanisms governing apple flower bud differentiation holds paramount importance for genetic enhancement and regulatory technology advancement in this domain [7].

The C2H2 zinc-finger transcription factor (TF), belonging to the Cys2/His2-type zinc finger protein family, harbors a distinctive QALGGH domain that is pivotal for DNA binding, thereby endowing it with regulatory roles in various plant physiological processes [8]. C2H2 zinc-finger TFs are ubiquitously distributed across plant species, with *Arabidopsis* possessing 176 members, while tobacco has 118 [9], tomato, 92 [10], maize, 211 [11], and rice, 189 [12]. *AtSUP* serves as an archetypal C2H2 zinc-finger TF that is prominently expressed in floral organ cells during the third stage of flower development in *Arabidopsis thaliana* [13]. In cabbage, approximately 76.9% of C2H2 zinc-finger TFs exhibit floral expression patterns, with the dysregulation of *BrDAZ3* expression resulting in aberrant pollen development [14]. In rice, *OsLRG1*, encoding a C2H2 zinc-finger TF, demonstrates predominant expression in spikelets and significantly influences spikelet organogenesis and grain-size regulation [15]. During flowering induction, C2H2, acting as a histone methyltransferase, modulates the *FLC* chromatin states, thereby modulating *FLC* activation or repression and participating in the FRI-dependent, vernalization, and autonomous flowering pathways [16]. Additionally, C2H2 zinc-finger TFs impact floral development by modulating hormone homeostasis [17]. For instance, the introduction of the *AtGIS* gene from *Arabidopsis* into tobacco mediates glandular trichome development in tobacco via GA signaling pathway regulation [18]. In summation, C2H2 zinc-finger TFs govern plant growth and development by fine-tuning hormone levels and orchestrating floral development processes.

The DELLA protein (aspartate-glutamate-leucine-leucine-alanine) stands as a well-known plant-specific transcriptional regulator that modulates gibberellin (GA) signaling, serving as a core regulatory factor in flowering. DELLA primarily modulates plant flowering by mediating target gene expression or interacting with other proteins. Presently, five member of the DELLA gene family have been identified in *Arabidopsis* [19]: three in tobacco, five in cabbage, one (*SLR1*) in rice [20], two (*DEARF8* and *DEARF9*) in maize, and one (*PROCERA*) in tomato [21]. Remarkably, various members of this family assume pivotal roles in flower development. For instance, *BcRGL1* assumes a central role in early shoot differentiation across diverse cabbage varieties, influencing branching and flowering [22]. In *Arabidopsis*, the deletion mutant gai-3 markedly retards the growth of floral organs, particularly affecting anther development and culminating in male sterility [23]. The initial identification of the DELLA family member in *Arabidopsis* unveiled the mutation of the gai-1 gene, resulting in dark green leaves, diminished growth vigor, and delayed flowering [24]. The CO and DELLA proteins represent pivotal constituents of the light and GA signaling pathways, respectively, modulating flowering time under long-day conditions within the leaf vascular system [25]. GA facilitates flowering by alleviating the DELLA-mediated repression of key flowering genes such as *LFY* and *SOC1* [26]. Simultaneously, DELLA interacts with the FKF1 protein to facilitate plant flowering by promoting DELLA protein degradation [27].

The F-box protein FLAVIN-BINDING KELCH REPEAT F-BOX 1 (FKF1) participates in light signaling in plants, serving as a blue-light receptor. Presently, the *FKF1* gene family encompasses five genes in mōsō bamboo, two in maize, and six in soybean [28]. Beyond the elucidated regulatory role of *FKF1* in *Arabidopsis*, *OsFKF1* in rice exhibits functional parallels but divergent roles: studies indicate that *OsFKF1* mutants exhibit delayed flowering under short-day, long-day, and natural long-day conditions [29]. In maize, the comparative analysis of flowering times demonstrates that the flowering time of three T2 homozygous mutants edited by *ZmFKF1* significantly lags behind that of the wild-type B104 [30]. The overexpression of *GmFKF1* in soybeans results in delayed flowering under long-day conditions [31]. *OsFKF1* upregulates the expression of the flower activator Ehd2 while downregulating the expression of the flower-repressive protein Ghd7. These regulatory factors modulate the expression of *Ehd1* in an up- and down-regulatory manner, respectively [30]. *FKF1* also orchestrates the degradation of the CYCLING DOF FACTOR 1 (CDF1) protein via the 26S proteasome pathway during the afternoon, thereby relieving CDF1’s inhibitory effect on *CO* transcription and inducing *FT* expression to promote flowering [32]. Additionally, FKF1 interacts with CONSTITUTIVE PHOTOMORPHOGENIC 1 (COP1) in a light-dependent manner, inhibiting its homodimerization and activity, consequently preventing the COP1-mediated degradation of CO proteins and thereby promoting flowering [33].

In conclusion, while the involvement of the *C2H2*, *DELLA*, and *FKF1* gene families in flowering regulation is well-documented in various plant species, their roles in apple flowering remain largely unexplored. Hence, this study employed bioinformatics methodologies to identify members of the *C2H2*, *DELLA*, and *FKF1* gene families in apples, aiming to investigate their structural characteristics, conserved domains, tissue-specific expression profiles, phylogenetic relationships, chromosomal distribution patterns, physicochemical properties of proteins, and motif predictions. Furthermore, we analyzed the expression patterns and subcellular localization of these gene family members, validated their expression dynamics during apple flowering induction using qRT-PCR, and conducted yeast two-hybrid screenings to identify their interacting proteins. These comprehensive analyses aim to establish a theoretical framework for further elucidating the functional roles of the *C2H2*, *DELLA*, and *FKF1* gene families in regulating apple flowering.

## 2. Results

### 2.1. Identification of C2H2, DELLA, and FKF1 Gene Families in the Apple Genome

We used HMM search software to detect genes from the C2H2, DELLA, and FKF1 families. Finally, the apple genome proved to contain 24 *C2H2* genes, 6 *DELLA* genes, and 6 *FKF1* genes. The apple *C2H2*, *DELLA*, and *FKF1* genes were sequentially named *MdC2H2-1*~*MdC2H2-24*, *MdDELLA1*~*MdDELLA6*, and *MdFKF1-1*~*MdFKF1-6*, based on their positions in the chromosomes. The analysis of physical and chemical properties showed (see Table 1) that the lengths of *C2H2*, *DELLA*, and FKF1 genes in the apple genome were 1076 to 6347 bp, 1640 to 1947 bp, and 2898 to 6558 bp, respectively. The number of amino acids ranged from 294 to 600, from 546 to 639, and from 371 to 632, and the molecular weights ranged from 34,436.24 to 64,123.49 aa, from 59,826.8 to 70,235.28 aa, and from 41,108.05 to 70,550.83 aa, among which the smallest molecular weights of the three gene families were *MdC2H2-23* (34,436.24 aa), *MdDELLA1* (59,826.8 aa), and *MdFKF1-2* (41,108.05 aa). Although *MdFKF1-4* and *MdDELLA1* were the shortest, their molecular weights were not the smallest, probably due to their large number of exons. The isoelectric points ranged from 5.73 to 9.29, from 4.95 to 5.63, and from 5.29 to 9.47. A prediction of hydrophobicity using ExPasy showed that the mean hydrophilicity values were all less than 0, and all three family members were hydrophilic proteins. Subcellular localization prediction found that most of the three families’ gene action sites in apples are located in the nucleus, and only *MdC2H2-11*, *MdC2H2-18*, *MdFKF1-1*, and *MdFKF1-5* are located in the cytoplasm.

### 2.2. Phylogenetic Analysis of the C2H2, DELLA, and FKF1 Gene Families

In this study, a multispecies phylogenetic tree was constructed by extracting the C2H2, DELLA, and FKF1 amino acid sequences from the genomes of *Arabidopsis* and rice and merging them with the MdC2H2, MdDELLA, and MdFKF1 sequences from apples. The results showed that the C2H2 family was divided into three groups (Figure 1A), and the MdC2H2 sequences were all clustered in groups II and III. No apple genes appeared in group I, only rice and *Arabidopsis* genes, indicating that there was a greater degree of differentiation between apples and *Arabidopsis* and rice that arose in the long-term evolutionary process, leading to the specificity of MdC2H2. There were 5 MdC2H2 sequences in group II and 19 MDC2H2 sequences in group III. Multiple MdC2H2 sequences were located in the same terminal evolutionary branch, suggesting that species-specific amplification of this gene family occurred in apples. The DELLA family was also divided into three groups (Figure 1B), with MdDELLA clustered in groups I and III. There were only two MdDELLA sequences and no other genes in group I and there were only two OsDELLA sequences in group II, suggesting that these two species had a greater degree of differentiation with *Arabidopsis* in the long-term evolutionary process, and that species-specific amplification of the DELLA family occurred. There were five AtDELLA and four MdDELLA sequences in group III. The FKF1 family was similarly divided into three groups (Figure 1C), with the MdFKF1 sequences all distributed in groups I and II and all AtFKF1 sequences in group III. The MdFKF1-1 sequence in group I and MdFKF1-4 sequence in group II appeared on separate branches, and the groups were directly homologous genes to each other. Based on the evolutionary mapping relationships and branch lengths, the three families of MdC2H2, MdDELLA, and MdFKF1 in apples showed some homology with *Arabidopsis* and rice, while the apple *C2H2*, *DELLA*, and *FKF1* genes were distributed in most of the groups, suggesting that the differentiation of the three families in apples was complete.

### 2.3. Gene Structure and Conserved Protein Motif Analysis of the C2H2, DELLA, and FKF1 Gene Families

Gene structure and conserved motif diversity were analyzed in all three gene families. Further insights into the evolutionary features of the MdC2H2, MdDELLA, and MdFKF1 families were gained by studying the structures of the gene body and conserved motifs of the amino acid sequence. The results showed that *MdC2H2-5* and *MdC2H2-21* contained only a CDS (translational region), which is consistent with the result that *MdC2H2-21* did not have the smallest molecular weight, although it had the shortest gene. Gene structure analysis revealed that the number of exons in the *MdC2H2* genome ranged from 1 to 10, with a minimum of only 1 exon in *MdC2H2-21* and a maximum of 10 exons in *MdC2H2-1*. Structural differences between the genes suggested that apple *MdC2H2* genes were not derived from simple duplication. In total, 10 conserved motifs were identified in the apple *MdC2H2* ZFP family, with 16 genes sharing a common motif, suggesting that these motifs are important components of the MdC2H2 protein sequence (Figure 2A). The *MdDELLA* family had no introns, only exons, and each gene contained only one oversized exon. Each gene contained the 10 identified motifs, and the alignment positions were basically the same, indicating that these 6 genes have high similarity (Figure 2B). The six *MdFKF1* genes had both CDSs and UTRs. The number of introns was 1~3. Ten conserved motifs were identified in the MdFKF1 family, of which the MdFKF1-3, MdFKF1-4, and MdFKF1-6 proteins all contained the ten motifs in essentially the same positions, suggesting that there was a high degree of similarity among these three genes. In contrast, MdFKF1-1, MdFKF1-2, and MdFKF1-5 contained only four to five motifs, suggesting that these three genes differed from the others (Figure 2C).

### 2.4. Chromosomal Distribution and Covariance Analysis of C2H2, DELLA, and FKF1 Genes in Apples

Chromosomal localization analysis showed that the 24 apple MdC2H2 family members were distributed on twelve chromosomes (Figure 3A). Chr01, Chr05, Chr09, Chr10, and Chr16 did not have *MdC2H2* genes. The *MdC2H2* genes were distributed on the remaining chromosomes, and six chromosomes had one *MdC2H2* gene. The six apple MdDEDLLA family members were distributed on six chromosomes (Figure 3B). Among them, the *MdDEDLLA* genes were distributed on all six chromosomes with one on each chromosome. The six apple *MdFKF1* family members were distributed on three chromosomes (Figure 3C), of which *MdFKF1-1* was located on the upper end of Chr03, *MdFKF1-2* and *MdFKF1-3* were located in the middle of Chr13, and *MdFKF1-4~MdFKF1-6* were located on Chr16, where *MdFKF1-4* was located on the upper end of the chromosome and *MdFKF1-5* and *MdFKF1-6* were located on the lower end of the chromosome.

In order to further study the evolutionary relationships between different species in greater depth and to reveal the origin, evolution, and function of the C2H2, DELLA, and FKF1 families, this study analyzed the collinearity between the genomes of *Arabidopsis*, apples, and rice (Figure 4A–C). The results showed that there were directly homologous *C2H2*, *DELLA*, and *FKF1* gene pairs existing between *Arabidopsis*, apples, and rice, with the number of homologous gene pairs between apples and *Arabidopsis* being greater than that between apples and rice.

### 2.5. Expression Profiles of C2H2, DELLA, and FKF1 Genes in Apple Flower Buds

To further investigate the role of the apple C2H2, DELLA, and FKF1 families in plant flowering, 19 *C2H2* genes, 6 *DELLA* genes, and 6 *FKF1* genes were selected from the dormant (S1) and expanding (S2) phases of flower buds and were analyzed using qRT-PCR. The fluorescence quantification results showed that the expression of genes in the MdC2H2 ZFP family was not high in the S1 period (Figure 5A), and only the expression of *MdC2H2-1*, *MdC2H2-2*, *MdC2H2-9*, and *MdC2H2-18* differed significantly from that in the S1 period in the S2 period (twice as much as that in the S1 period). The expression of the other genes did not differ significantly from that in the S1 period. In the MdDELLA family (Figure 5B), the expression of *MdDELLA1*, *MdDELLA2*, and *MdDELLA4* in the S1 period was significantly different from and higher than that in the S2 period, and the expression of *MdDELLA3*, *MdDELLA5*, and *MdDELLA6* in the S1 period was not significantly different from that in the S2 period. In the *MdFKF1* family (Figure 5C), *MdFKF1-1*, *MdFKF1-2*, *MdFKF1-3*, and *MdFKF1-5* had almost zero expression in S1 and S2, and the expression of *MdFKF1-4* and *MdFKF1-6* was very low in S1, with the difference in the expression of *MdFKF1-4* between S1 and S2 being obvious as a more than 10-fold increase.

### 2.6. Subcellular Localization of Apple C2H2, DELLA, and FKF1 Proteins

The higher expression levels of the *MdC2H2-18*, *MdDELLA1*, and *MdFKF1-4* genes for the two periods of apple flower buds suggest that they may have some regulatory roles in apple flower development. Therefore, subcellular localization analysis was conducted on the MdC2H2-18, MdDEALL1, and MdFKF1-4 proteins. The transient expression of CAM-EGFP-MdC2H2-18, CAM-EGFP-MdDEALL1, and CAM-EGFP-MdFKF1-4 in tobacco by agrobacterium further confirmed the strong fluorescence that mainly appeared in the cell nucleus (Figure 6), indicating the localization of these three genes in the nucleus and their involvement as nuclear proteins in regulating apple flowering.

### 2.7. Yeast Two-Hybrid Validation

Through the previous experiments, we gained a basic understanding of the spatial expression sites of *MdC2H2-18*, *MdDELL1A*, and *MdFKF1-4* in plants and the biological functions of these genes. To further explore which proteins interact with each other and affect their functions when these three proteins are active, we performed yeast two-hybrid experiments on these three proteins. First, the successfully constructed pGBKT7-MdDELLA1 decoy vector was co-transfected into the Y2Hgold yeast sensory state with an empty pGBKT7. pGBKT7-VP16 and pGBKT7 were used as positive and negative controls and were inverted and incubated at 30 °C for 3 days. It was found that all three co-transformed yeasts grew white spots in two-deficient medium (Figure 7), proving that the plasmid co-transformation was successful. The pGBKT7-MdDELLA1 co-transformed with pGBKT7 and the negative control did not grow spots on four-deficient medium, while the positive control turned blue in four-deficient medium, showing that the apple *MdDELLA1* gene did not exhibit auto-activating activity.

In order to test whether MdDELLA1 and MdC2H2-18, and MdDELLA1 and MdFKF1-4, have a reciprocal relationship, the successfully constructed pGBKT7-MdDELLA1 vector was co-transfected into the Y2Hgold yeast sensory state, together with the pGBKT7-MdC2H2-18 vector, and the pGBKT7-MdDELLA1 vector was co-transfected into the Y2Hgold yeast sensory state with the pGBKT7-MdFKF1-4 vector. Yeasts co-transformed with pGADT7-T and pGBKT7-53 were used as a positive control, and yeasts co-transformed with pGADT7-T and pGBKT7-lam were used as a negative control. As can be seen from the results of the yeast hybrids (Figure 8), all co-transformed yeasts grew white spots on the double-deficient medium, which proved that the plasmid co-transformation was successful. Transferring yeast grown on two-deficient medium to four-deficient and four-deficient+ medium for cultivation revealed that the positive control and the MdDELLA1 that were co-transformed with the MdC2H2-18 groups of yeasts were able to grow normally and turn blue, while the negative control and the MdDELLA1 that were co-transformed with MdFKF1-4 groups were not able to grow on the medium. The results showed that MdDELLA1 and MdC2H2-18 interacted, and it was hypothesized that a complex of these two genes might be involved in the regulation of floral organ development, whereas MdDELLA1 did not have any direct interactions with MdFKF1-4.

## 3. Materials and Methods

### 3.1. Test Materials

The test materials used in this study were from apple trees (*Malus domestica* Borkh. cv. Red Fuji) and were provided by the Institute of Horticultural Crops, Xinjiang Academy of Agricultural Sciences. All fruit trees were planted throughout the year at the Apple Experimental Station in Yecheng County, Kashgar Prefecture, Xinjiang Province. Flower buds of uniform size were collected from 3 10-year-old apple trees during the dormant and expansion periods, and 6-8 flower buds were collected from each tree in each period.

### 3.2. Identification and Prediction of Physicochemical Properties of C2H2, DELLA, and FKF1 Genes in Apples

Genome-wide apple data were downloaded from the GDR database (https://www.rosaceae.org/, accessed on 28 April 2023). The local BLAST database was constructed with apple protein sequences. The structural domains of C2H2, DELLA, and FKF1 in the Pfam database (https://pfam.xfam.org, accessed on 28 April 2023) were used as a model, and protein sequences containing these three structural domains were screened using the HMM program to obtain candidate genes for the C2H2, DELLA, and FKF1 of the apple genome, respectively [34]. The candidate genes, *C2H2*, *DELLA*, and *FKF1*, were analyzed using structural domains from three major databases (SMART https://smart.embl.de/, accessed on 28 April 2023), CDD-search (https://www.ncbi.nlm.nih.gov/cdd/, accessed on 28 April 2023), and PFAM (https://pfam.xfam.org/, accessed on 28 April 2023)). Genes that did not contain typical structural domains were deleted, and apple C2H2, DELLA, and FKF1 gene family members were obtained. The physicochemical properties of the candidate protein sequences of transcription factors of *C2H2*, *DELLA*, and *FKF1* genes were predicted using the online analysis software Expasy (https://www.expasy.org/tools/, accessed on 28 April 2023), and secondary structure and subcellular localization predictions were performed using the Prabi website (https://npsa-prabi.ibcp.fr/cgi-bin/npsa_automat.pl?page=/NPSA/npsa_sopma.html, accessed on 28 April 2023).

### 3.3. Gene Structure, Analysis, and Phylogenetic Tree Construction

Multiple sequence alignments of the identified apple C2H2, DELLA, and FKF1 family members were performed using the DNAMAN software. The MEME website (https://meme-suite.org/tools/meme/, accessed on 19 May 2023) was used to predict their preservation motifs (the number of functional domains was set to 10), and the gene structures were mapped using the TBtools software. The phylogenetic tree was constructed using the neighbor-joining function in the MEGA7.0 software [35]. The Poisson distribution model was set, and the bootstrap value was modified to 2000, then trees were constructed for the candidate gene sequences of apple C2H2, DELLA, and FKF1.

### 3.4. Chromosome Distribution and Co-Linearity Analysis

The TBtools software was used to extract the C2H2, DELLA, and FKF1 gene position information from the apple genome files with gene annotation files and to map the positions of the *C2H2*, *DELLA*, and *FKF1* genes on the chromosomes [36]. The McscanX function in the TBtools software was used to extract the apple C2H2, DELLA, and FKF1 gene families containing covariance relationships, and covariance analyses between apples and different species of *Arabidopsis thaliana* and rice were plotted.

### 3.5. Quantitative Fluorescence Analysis

The total RNA was extracted using the RNA-prep Pure Plant Kit (DP441) from Tiangen Biochemical Technology (Beijing) Co. (Beijing, China). The qRT-PCR system (20 μL) contained 10 μL 2× ChamQ Universal SYBR qPCR Master Mix (Novozymes, Franklinton, NC, USA), 8.2 μL ddH_2_O, 0.4 μL each of upstream- and downstream-specific primers, and 1 μL of cDNA template. Using a Roche Light Cycler 96 real-time quantitative fluorescence PCR instrument, the reaction procedure involved 120 s of pre-denaturation at 94 °C, 5 s of denaturation at 94 °C, 15 s of annealing time, and 10 s of extension at 72 °C for 45 cycles. The relative expression was calculated using three biological replicates and the 2^−ΔΔCT^ method, using *MdMDH* as the internal reference gene, and the relative expression was the relative value of the treated and control groups [37]. Fluorescent quantitative PCR primers were designed based on the conserved sequences of apple C2H2, DELLA, and FKF1 gene family members (Schedule 1).

### 3.6. Subcellular Localization

The sequences of *MdC2H2-18*, *MdDELLA1*, and *MdFKF1-4* were constructed on the pGBKT7 vector to obtain the MdC2H2-18-GFP, MdDELLA1-GFP, and MdFKF1-4-GFP fusion proteins, respectively. Tobacco leaves were injected at 4 weeks of age, and fluorescence was observed 48 h later. Imaging was performed using an FV3000 confocal laser scanning microscope (Olympus, Japan In Olympus Corporation Global Headquarters 2951 Ishikawa-machi, Hachioji-shi, Tokyo 192-8507, Japan).

### 3.7. Yeast Hybridization

The coding regions of *MdC2H2-18*, *MdDELLA1*, and *MdFKF1-4* were cloned into the pGBKT7 vector. The marker proteins used were nuclear localization marker proteins. Specific vector combinations were transformed in the Y2HGold strain. Positive transformants grown on an SD-Trp/-Leu (DDO) medium were inoculated onto screening media (SD/-Trp/-Leu/-His/X-a-gall, SD/-Trp/-Leu/-His/-Ade/X-a-gall) to identify possible interactions.

## 4. Discussion

Flowering constitutes a pivotal stage in the plant life cycle, wherein the precise timing of this event holds paramount importance for ecological adaptation, survival, and reproductive success [38]. Previous research has extensively documented the involvement of numerous genes in flowering regulation across various plant species. Notably, C2H2 zinc-finger TFs, DELLA proteins, and FKF1 genes have emerged as key regulators of flowering in species such as tobacco, rice, and cabbage. Hence, this study aims to analyze these three families of flowering genes in apples using bioinformatics methodologies.

In this investigation, we identified 24 apple MdC2H2 zinc-finger TFs. Analysis of the gene structure revealed a variable number of introns in the MdC2H2, ranging from 1 to 7, a higher count compared to crops like tomatoes and oilseed rape [38]. This observation suggests a potentially heightened frequency of recombination among *MdC2H2* zinc-finger TFs, potentially facilitating the evolutionary trajectory of the apple species with corresponding regulatory implications. Phylogenetic tree analyses demonstrated that C2H2 zinc-finger TFs within the same clade exhibit considerable homology, with duplicated genes likely originating from a common ancestor and potentially harboring similar functions. Gene duplication stands as a significant driver behind the rapid expansion and evolution of gene families [39]. Notably, it is evident that among the 19 *MdC2H2* zinc-finger TFs assessed for fluorescence quantification (Figure 5A), 12 exhibited higher expression levels during the flower bud expansion phase (S2) compared to the flower bud dormancy phase (S1), implying their putative role in promoting flowering.

We identified an additional six apple *MdDELLA* sequences. Gene structure analysis revealed a notable absence of introns within any of the *MdDELLA* sequences, which is consistent with findings pertaining to soybeans. It has been demonstrated that genes devoid of or possessing few introns exhibit rapid expression in response to both biotic and abiotic stresses [40,41]. Subcellular localization assays indicated nuclear localization for all MdDELLA proteins, aligning with the subcellular localization patterns observed in homologous proteins across species such as cucurbits, tobacco, and grapes [42,43,44]. Transient transformation experiments conducted on tobacco leaflets reaffirmed the presence of nuclear localization for the MdDELLA1 protein in apples. Notably, it was observed that the expression levels of five out of the six MdDELLA genes during the bud dormancy period (S1) exceeded those during the bud expansion period (S2), with *MdDELLA1*, *MdDELLA2*, and *MdDELLA4* exhibiting the most pronounced differences (Figure 5B). This expression pattern parallels the CCCH zinc-finger TFs in Japanese apricot (*Prunus mume* Sieb.) [45], suggesting a potential association between *MdDELLA* genes and the induction of apple flower buds to alleviate dormancy.

Furthermore, we identified six apple *MdFKF1* genes; analysis of the gene structure revealed variable intron counts ranging from one to three. However, intronless F-box genes are more prevalent in other reported plant species. For instance, 45% of F-box genes in *Arabidopsis* were predicted to lack introns [46], along with 40.76% in rice [47] and over 40% in maize [29]. This observation implies a divergent evolutionary pattern for *MdFKF1* compared to other plant species. It was evident that only *MdFKF1*-4 among the six *MdFKF1* genes exhibited the highest expression level during the bud expansion stage (Figure 5C), with a 20-fold increase compared to the bud dormancy stage, while the remaining genes displayed relatively low expression levels with insignificant differences. Thus, it is conjectured that *MdFKF1-4* may possess the most typical sequence and conserved function within the MdFKF1 family, playing a pivotal role in apple flower development as an indispensable member of the apple FKF1 family.

Gene expression typically exhibits temporal and spatial specificity and is closely correlated with function. In cabbage, C2H2-type zinc-finger TFs are expressed across various tissues, including the roots, stems, leaves, and flowers, with 76.9% of genes expressed in the flowers [48]. Moreover, the tissue-specific expression of DELLA genes exists within different varieties of the same crop. For instance, *MeGRAS33* transcript abundance was higher in the roots of cassava W14 but was lower in the roots of Arg7 and KU50 [49]. Similarly, *AtFKF1* was expressed specifically in different floral organs of *Arabidopsis thaliana*, promoting early flowering [50]. In this study, qPCR was utilized to ascertain the expression patterns of three apple gene families across two flower bud stages. The results revealed the significant differential expression of *MdC2H2-18*, *MdFKF1-4*, and *MdDELLA1* between the two flower bud stages, warranting further investigation into their potential interactions and subsequent effects on flowering.

As a transcriptional repressor within the gibberellin (GA) signal transduction pathway, the DELLA protein assumes widespread involvement in plant growth and development. Positioned within the nucleus, DELLA proteins serve as growth inhibitory factors that are capable of direct interaction with key transcription factors in plants, thereby modulating plant growth and development [51]. In *Arabidopsis thaliana* studies, FKF1 was discovered to promote flowering by negatively modulating the stability of DELLA proteins through the formation of a protein complex with DELLA proteins [32]. However, such interactions have seldom been reported in apples. Thus, in this investigation, yeast two-hybrid experiments were conducted on the three proteins MdC2H2-18, MdFKF1-4, and MdDELLA1 to ascertain whether a reciprocal relationship exists among them, thereby potentially regulating flowering in apples. The experimental findings revealed a reciprocal association between *MdDELLA1* and *MdC2H2-18*, indicating that their interaction may indeed govern apple flowering regulation. Conversely, no reciprocal relationship was observed between *MdDELLA1* and *MdFKF1-4*, which is a discrepancy compared to *Arabidopsis thaliana* studies, one that is likely attributable to the divergent roles played by these genes across different crops.

## 5. Conclusions

We identified 24 genes from the C2H2 family, 6 genes from the DELLA family, and 6 genes from the FKF1 family in the apple genome with high confidence. Phylogenetic analyses classified the genes within each family into three distinct subgroups, with various aspects of protein physiological properties, protein tertiary structures, and conserved motifs being contingent. Repetitive events within the apple genome were delineated through the homologs inferred. qRT-PCR analysis revealed notable disparities in the expression levels of *MdC2H2-18*, *MdDELLA1*, and *MdFKF1-4* during apple bud development. Subsequent yeast two-hybrid analysis unveiled an interaction between MdC2H2-18 and MdDELLA1. The comprehensive genome-wide identification of C2H2, DELLA, and the FKF1 gene family in apples enhances our understanding of the molecular mechanisms underlying apple bud development.

## Figures and Tables

**Figure 1 ijms-25-07510-f001:**
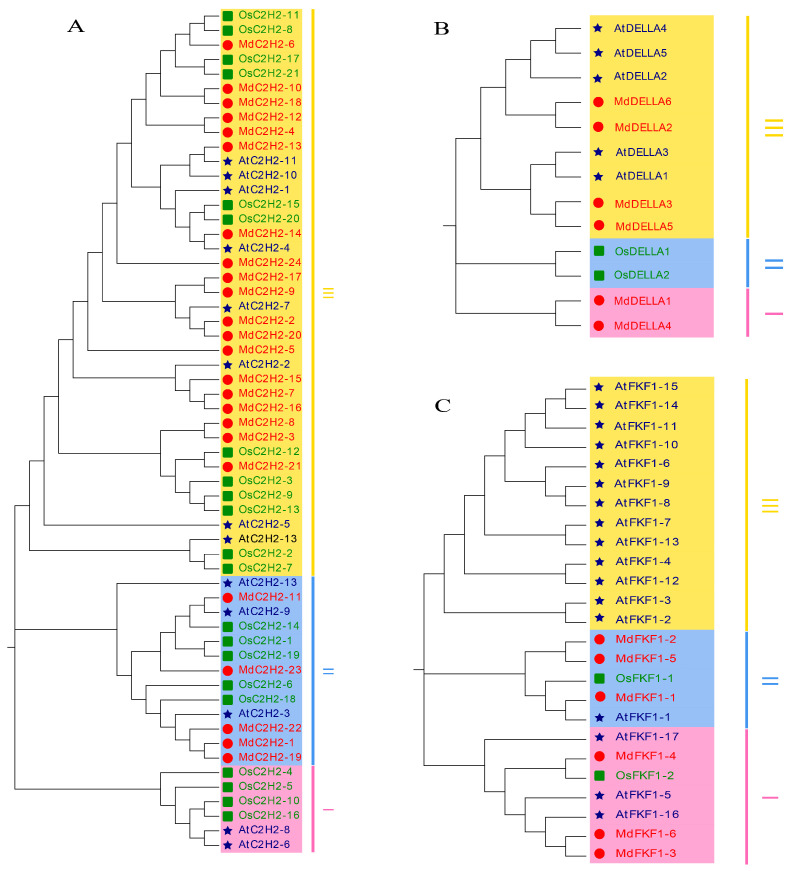
Phylogenetic relationships of three gene families in *Arabidopsis* (blue star), rice (green square), and apple (red circle) (**A**) C2H2; (**B**) DELLA; (**C**) FKF1. The position of I, II, and III lines indicates that the families are divided into three subgroups.

**Figure 2 ijms-25-07510-f002:**
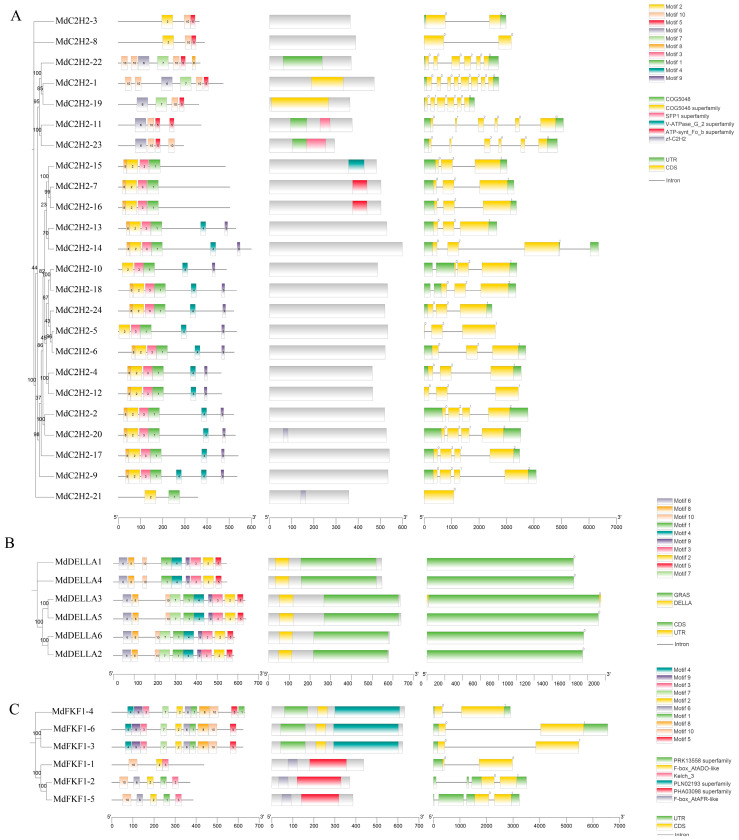
Gene structure and conserved motif analysis of three gene families in the apple genome: (**A**) C2H2; (**B**) DELLA; (**C**) FKF1.

**Figure 3 ijms-25-07510-f003:**
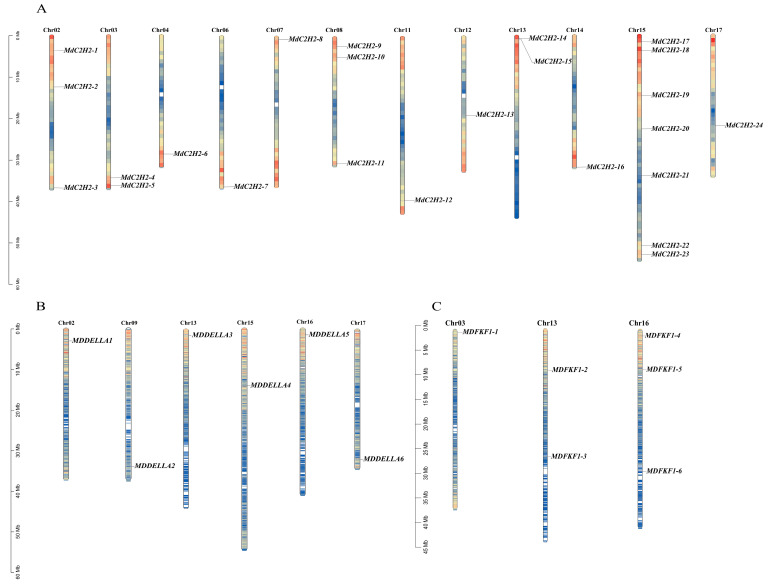
Chromosomal localization analysis of three gene families in the apple genome: (**A**) C2H2; (**B**) DELLA; (**C**) FKF1.

**Figure 4 ijms-25-07510-f004:**
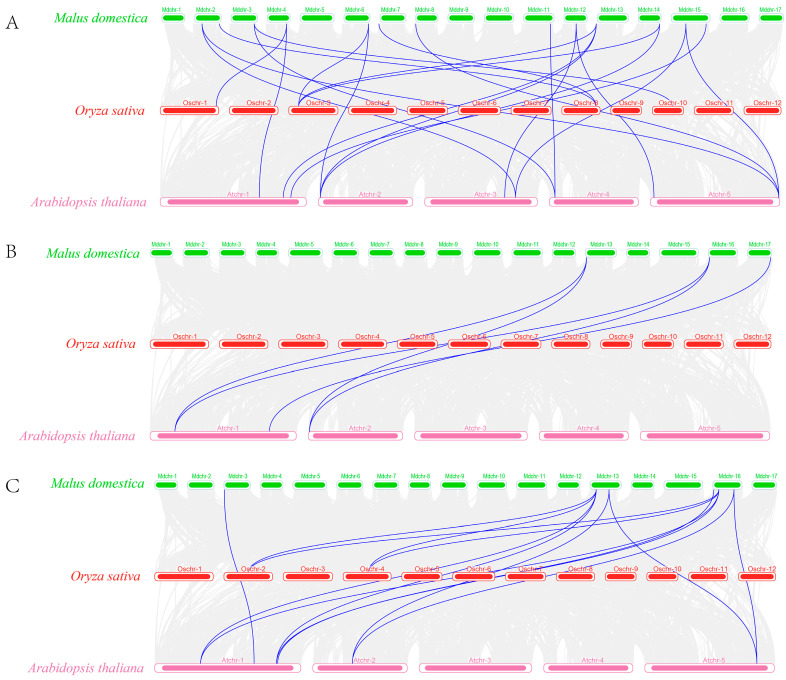
Covariance analysis of three gene families in apple, *Arabidopsis*, and rice: (**A**) C2H2; (**B**) DELLA; (**C**) FKF1. Blue lines indicate the covariance of the three gene families in apple and rice and in apple and Arabidopsis.

**Figure 5 ijms-25-07510-f005:**
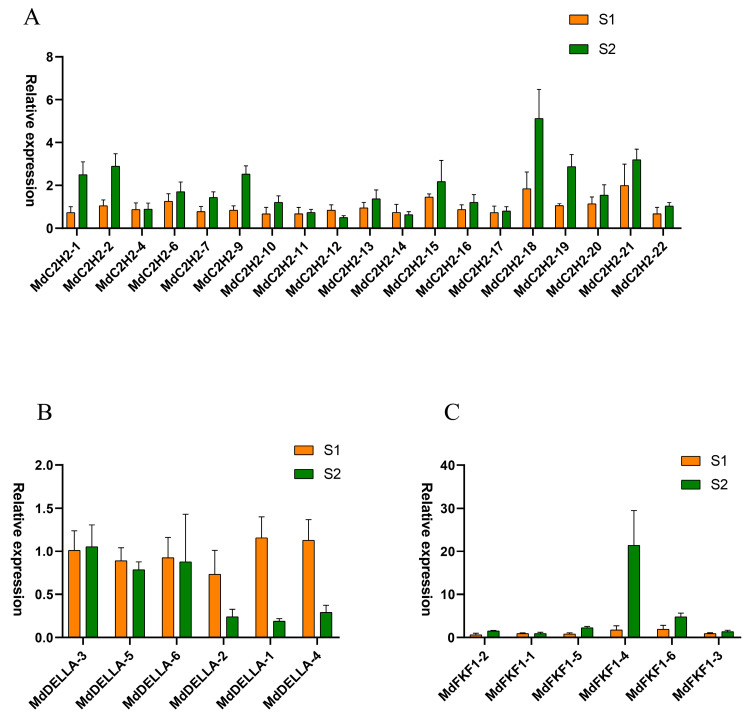
Expression of three gene families in apple flower buds at two time periods: (**A**) C2H2; (**B**) DELLA; (**C**) FKF1. S1: dormancy period of flower buds; S2: the expansion phase of flower buds.

**Figure 6 ijms-25-07510-f006:**
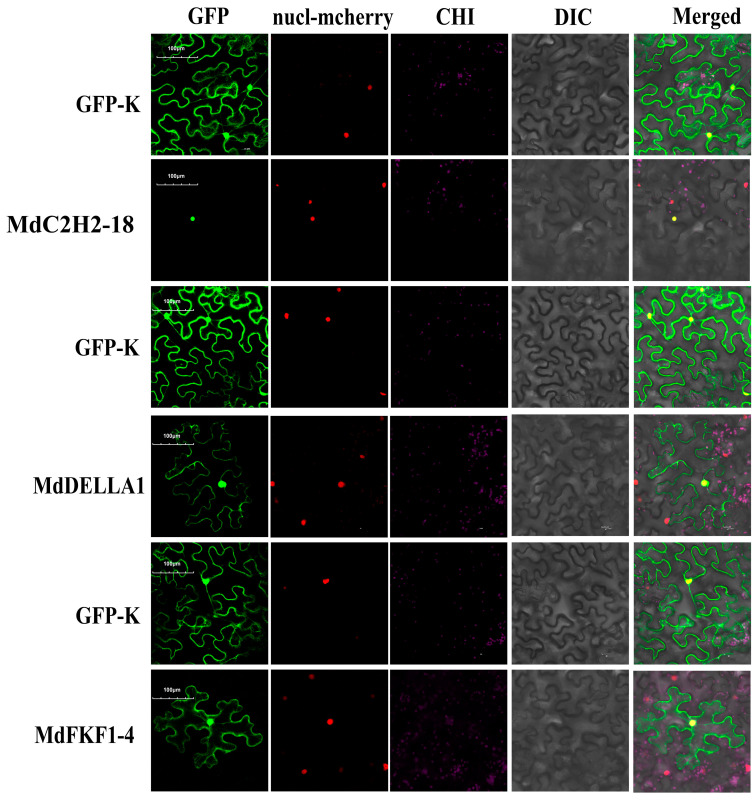
Subcellular localization of the fusion proteins MdC2H2-18-GFP, MdDELLA1-GFP, and MdFKF1-4-GFP in *N. benthamiana* leaves. GFP: green fluorescent protein; CHI: chloroplast fluorescence channel; bright field: visible light; Merged: overlay of the bright field, green fluorescence, and red fluorescence images. Bar = 20 μm. Green fluorescence indicates that all three genes are localised in the nucleus.

**Figure 7 ijms-25-07510-f007:**
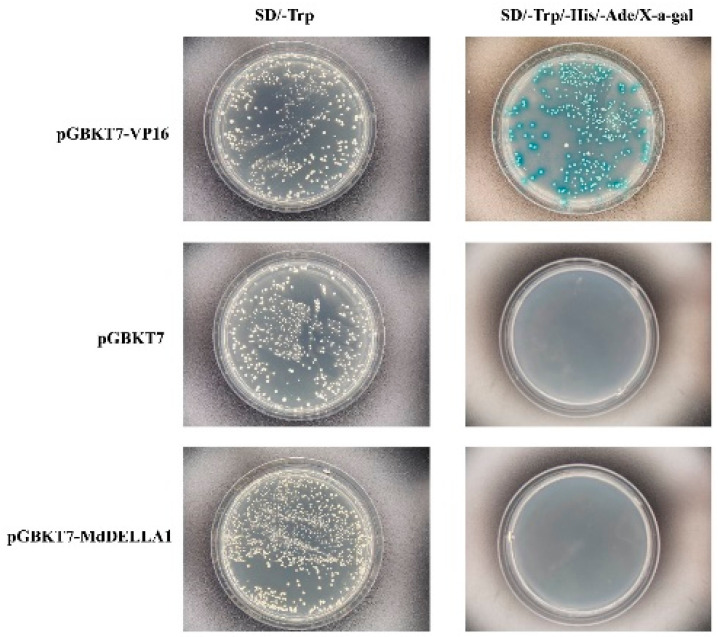
MdDELLA1 self-activation detection. The MdDELLA-1 protein is non-toxic and non-self-activating.

**Figure 8 ijms-25-07510-f008:**
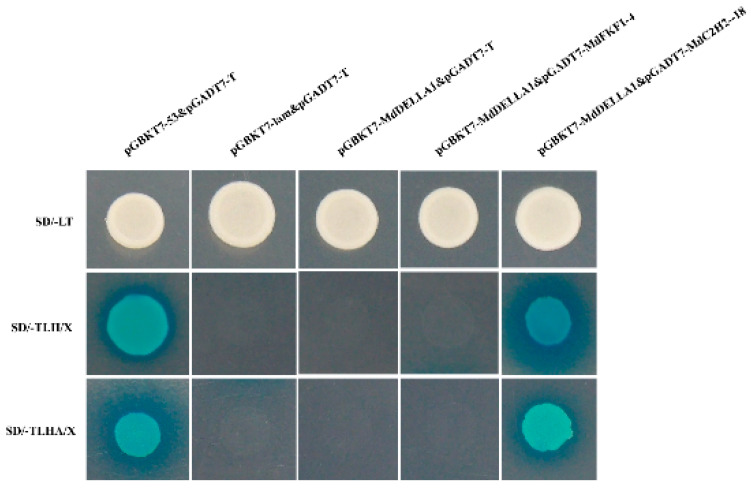
Yeast two-hybrid assay to detect whether MdDELLA-1 regulates downstream genes (MdFKF1-4 and MdC2H2-18). pGBKT7-53&pGADT7-T was used as the negative control; pGBKT7-lam&pGADT7-T was used as the positive control.

**Table 1 ijms-25-07510-t001:** Basic information of amino acid sequences encoded by the members of the C2H2, DELLA, and FKF1 gene families in apples.

Gene Name	Gene ID	Chromosome Location	Length of CDS (bp)	Physicochemical Properties				Subcellular Localization
				No.	Mw	pI	GRAVY	
MdC2H2-1	MD02G1048300	Chr2	2716	473	53,935.18	8.58	−0.785	nucl
MdC2H2-3	MD02G1313400	Chr2	2983	365	40,796.93	8.8	−0.608	nucl
MdC2H2-2	MD02G1151700	Chr2	3781	520	54,260.67	8.83	−0.495	nucl
MdC2H2-4	MD03G1258200	Chr3	3533	464	50,955.33	9.16	−0.637	nucl
MdC2H2-5	MD03G1283900	Chr3	2591	534	58,928.67	8.82	−0.818	nucl
MdC2H2-6	MD04G1203000	Chr4	3703	522	57,092.7	8.99	−0.746	nucl
MdC2H2 7	MD06G1234600	Chr6	3271	503	56,383.07	8.94	−0.884	nucl
MdC2H2 8	MD07G1008700	Chr7	3170	389	43,693.48	7.04	−0.797	nucl
MdC2H2-11	MD08G1239600	Chr8	5069	373	42,921.8	6.24	−1.145	cyto
MdC2H2-10	MD08G1063600	Chr8	3377	488	54,409.49	8.86	−0.18	nucl
MdC2H2-9	MD08G1033100	Chr8	4082	535	55,755.13	8.81	−0.437	nucl
MdC2H2-12	MD11G1278800	Chr11	3430	466	51,228.26	9.2	−0.691	nucl
MdC2H2-13	MD12G1120300	Chr12	2650	530	58,194.12	9.23	−0.666	nucl
MdC2H2-14	MD13G1013100	Chr13	6347	600	64,123.49	9.29	−0.717	nucl
MdC2H2-15	MD13G1015000	Chr13	3018	483	52,950.02	8.94	−0.784	nucl
MdC2H2-16	MD14G1241600	Chr14	3364	503	56,507.32	9.02	−0.881	nucl
MdC2H2-23	MD15G1431100	Chr15	4846	294	34,436.24	7.92	−0.773	nucl
MdC2H2-17	MD15G1029700	Chr15	3481	541	56,410.92	8.62	−0.445	nucl
MdC2H2-18	MD15G1058000	Chr15	3342	533	58,976.45	8.54	−0.798	cyto
MdC2H2-20	MD15G1266200	Chr15	3521	528	55,077.62	9.04	−0.481	nucl
MdC2H2-21	MD15G1323400	Chr15	1076	358	40,054.06	5.73	−0.636	nucl
MdC2H2-19	MD15G1186600	Chr15	1835	363	41,314.08	9.16	−0.684	nucl
MdC2H2-22	MD15G1411000	Chr15	2708	369	42,755.28	8.34	−0.932	nucl
MdC2H2-24	MD17G1186000	Chr17	2471	521	56,980.48	8.34	−0.668	nucl
MdDELLA3	MD13G1022100	Chr13	1947	635	69,734.67	5.3	−0.28	nucl
MdDELLA5	MD16G1023300	Chr16	1919	639	70,235.28	5.25	−0.287	nucl
MdDELLA6	MD17G1260700	Chr17	1754	584	63,697.47	4.95	−0.289	nucl
MdDELLA2	MD09G1264800	Chr09	1742	580	63,166.16	5.09	−0.215	chlo
MdDELLA1	MD02G1039600	Chr02	1640	546	59,826.8	5.63	−0.114	chlo
MdDELLA4	MD15G1180500	Chr15	1643	547	59,896.09	5.53	−0.091	nucl
MdFKF1-2	MD13G1117000	Chr13	3505	371	41,108.05	5.75	−0.138	nucl
MdFKF1-1	MD03G1007500	Chr3	2972	437	49,203.71	9.47	−0.243	cyto
MdFKF1-5	MD16G1117600	Chr16	3235	386	43,284.64	6.2	−0.21	cyto
MdFKF1-4	MD16G1018000	Chr16	2898	632	70,550.83	5.53	−0.351	nucl
MdFKF1-6	MD16G1255700	Chr16	6558	623	67,868.46	5.29	−0.112	nucl
MdFKF1-3	MD13G1249400	Chr13	5468	623	68,016.72	5.34	−0.107	nucl

Notes: No.: number of amino acids; Mw: molecular weight; pI: isoelectric point of amino acids.

## Data Availability

The data that support the findings of this study are available from the corresponding author upon request.

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
