# Peer review of "Identification of Apple Flower Development-Related Gene Families and Analysis of Transcriptional Regulation"

_ijms, 2024, doi:10.3390/ijms25147510_

Round 1
Reviewer 1 Report
Comments and Suggestions for Authors
The manuscript contains some obvious results about gene structure and phylogenetics, but prediction of cis-elements (TFBS motifs) should be completely redone based on the modern data sources and at least standard position weight matrix approach derived from whole genome DAP-seq or ChIP-seq studies. Yet, experimental result still promote the manuscript.
Line 22
…Cis-acting regulatory elements in the promoter suggested a role for MdC2H2, MdDELLA, and MdFKF1 proteins in responding to environmental stimuli
Why the term ‘transcription factor’ (TF) is not used in the text anywhere? Cis-acting regulatory elements suspect nucleotide context specificity
38-40
…There are now more established regulatory models for the pathways that regulate flowering in plants, and a large number of genes regulating flowering have been identified, such as C2H2, DELLA, and FKF1
Should be clearly indicated whether C2H2, DELLA, and FKF1 mean genes or gene families, see https://www.arabidopsis.org/
Refer to Plant-TFClass and PlantTFDB (PlantRegMap) for the recent correct classification of plant transcription factors, since below you often write about families or classes.
Blanc-Mathieu, R., Dumas, R., Turchi, L., Lucas, J., Parcy, F. (2023) Plant-TFClass: a structural classification for plant transcription factors. Trends Plant Sci. S1360-1385(23), 00227-3. 10.1016/j.tplants.2023.06.023
Tian, F., Yang, D.C., Meng, Y.Q., Jin, J., Gao, G. (2020) PlantRegMap: charting functional regulatory maps in plants. Nucleic Acids Res. 48(D1), D1104–D1113. 10.1093/nar/gkz1020
Note that supplement for PlantTFDB paper contains full list of plant TFs, verify that these three clades refer to TFs. (C2H2 and DELLA (GRAS) are real and putative TFs, what about FKF1 ?) You may read this paper to clearly catch the definition of TF
Lambert, S.A., Jolma, A., Campitelli, L.F., Das, P.K., Yin, Y., Albu, M., Chen, X., Taipale, J., Hughes, T.R., Weirauch, M.T. (2018) The Human transcription factors. Cell 172, 650–665.
116, 153 156 162 etc.
DEALL -> DELLA ??? or not
Rename this table. Subcellular localization and length are not physicochemical properties.
Table 1. Physicochemical properties of amino acid sequences encoded by members of the C2H2, DEALL, and FKF1 gene families in apples
Table is rather long, it fits the supplementary data. The text in lines 120-136 is too long, many details. E.g. … A prediction of subcellular localization revealed that most of the three families' gene action sites in apples were located in the nucleus, with only MdC2H2-11, MdC2H2-18, MdFKF1-1, and MdFKF1-5 located in the cytoplasm… is good.
Fig.1
Does tree for C2H2 zinc fingers in accordance with classification from Plant-TFClass, where C2H2 ZF class = C2H2 & IDD families ?
186
5’ or 3’ UTR ?
200
on 12 chromosomes -> on twelve chromosomes
(12th chromosome is not twelve chromosomes)
Fig.3 201 -215 = too simple and obvious text
It is not biology. It is too descriptive. What is meaning of the exact position of a gene in genome? I think it's almost irrelevant to the biological meaning.
226
…This suggests that apples have a more advanced evolutionary relationship with the C2H2, DEALL, and FKF1 gene families of Arabidopsis.
What is novelty of this result? E.g. I know that human and mouse are taxonomically closer than human and cat. Try to briefly discuss your result.
232
…start sites of the apple MdC2H2, MdDEALL, and MdFKF1 gene families
Transcription start sites of genes from the families...
Fig. 5 is completely wrong
Axis X must refer to certain transcription factors, in particular, recognition models for their binding sites
You should predict transcription factor binding sites (TFBS) motifs instead of certain ‘boxes’. Use FIMO https://meme-suite.org/meme/tools/fimo, apply TFBS motifs from JASPAR & CISBP, or Plant Cistrome
Rauluseviciute, I., Riudavets-Puig, R., Blanc-Mathieu, R., Castro-Mondragon, J. A., Ferenc, K., Kumar, V., Lemma, R. B., Lucas, J., Chèneby, J., Baranasic, D., et al., (2024) JASPAR 2024: 20th anniversary of the open-access database of transcription factor binding profiles. Nucleic Acids Res, 52(D1), D174-D182
Weirauch, M. T., Yang, A., Albu, M., Cote, A. G., Montenegro-Montero, A., Drewe, P., Najafabadi, H. S., Lambert, S. A., Mann, I., Cook, K., et al. (2014) Determination and inference of eukaryotic transcription factor sequence specificity. Cell 158, 1431–1443.
O'Malley, R.C., Huang, S.C., Song, L., Lewsey, M.G., Bartlett, A., Nery, J.R., Galli, M., Gallavotti, A., Ecker, J.R. (2016) Cistrome and epicistrome features shape the regulatory DNA landscape. Cell 165, 1280–1292.
I suspect that your boxes are something like degenerate consensus or too short matrices, they may have too high false positive rate, you may take all genes for Arabidopsis or apple and test whether any motif is specific for certain gene set compared to gene set of all genes. PlanCARE was published more than 20 years ago, its motifs are not based on modern NGS techniques like ChIP-seq and DAP-seq
Magali Lescot et al. PlantCARE, a database of plant cis-acting regulatory elements and a portal to tools for in silico analysis of promoter sequences Nucleic Acids Res. 2002 Jan 1;30(1):325-327.
Comments on the Quality of English Languagerelatively good, at least not awful.
Author Response
Dear Editors and Reviewer
Thank you for your valuable comments on our manuscript titled “Identification of apple flower development-related gene families and analysis of transcriptional regulation” (Manuscript ID: ijms-2976358). Firstly, we apologize for all the novice errors or formatting issues in our manuscript, we have revised the entire text to ensure its academic quality. Your suggestions have been very helpful in revising and improving our paper and have significant guidance for our research. We have carefully studied these comments and proofread and revised the entire text. The corresponding modifications are marked in red in the revised manuscript, and we hope these changes will be approved by you.
Q1. Line 22
…Cis-acting regulatory elements in the promoter suggested a role for MdC2H2, MdDELLA, and MdFKF1 proteins in responding to environmental stimuli
Why the term ‘transcription factor’ (TF) is not used in the text anywhere? Cis-acting regulatory elements suspect nucleotide context specificity
A1. We have eradicated the erroneous statement.
Q2. 38-40
…There are now more established regulatory models for the pathways that regulate flowering in plants, and a large number of genes regulating flowering have been identified, such as C2H2, DELLA, and FKF1
Should be clearly indicated whether C2H2, DELLA, and FKF1 mean genes or gene families, see https://www.arabidopsis.org/
A2. We have amended the relevant expression at line 48.
Q3. Refer to Plant-TFClass and PlantTFDB (PlantRegMap) for the recent correct classification of plant transcription factors, since below you often write about families or classes.
Blanc-Mathieu, R., Dumas, R., Turchi, L., Lucas, J., Parcy, F. (2023) Plant-TFClass: a structural classification for plant transcription factors. Trends Plant Sci. S1360-1385(23), 00227-3. 10.1016/j.tplants.2023.06.023
Tian, F., Yang, D.C., Meng, Y.Q., Jin, J., Gao, G. (2020) PlantRegMap: charting functional regulatory maps in plants. Nucleic Acids Res. 48(D1), D1104–D1113. 10.1093/nar/gkz1020
A3. We have proofread and rectified this issue throughout the manuscript, appreciating your suggestion.
Q4. Note that supplement for PlantTFDB paper contains full list of plant TFs, verify that these three clades refer to TFs. (C2H2 and DELLA (GRAS) are real and putative TFs, what about FKF1 ?) You may read this paper to clearly catch the definition of TF
Lambert, S.A., Jolma, A., Campitelli, L.F., Das, P.K., Yin, Y., Albu, M., Chen, X., Taipale, J., Hughes, T.R., Weirauch, M.T. (2018) The Human transcription factors. Cell 172, 650–665.
A4. FKF1 is a gene family
Q5. 116, 153 156 162 etc.
DEALL -> DELLA ??? or not
A5. DELLA is indeed correct, we appreciate your rectification and have revised it accordingly.
Q6. Rename this table. Subcellular localization and length are not physicochemical properties.
Table 1. Physicochemical properties of amino acid sequences encoded by members of the C2H2, DEALL, and FKF1 gene families in apples
Table is rather long, it fits the supplementary data. The text in lines 120-136 is too long, many details. E.g. … A prediction of subcellular localization revealed that most of the three families' gene action sites in apples were located in the nucleus, with only MdC2H2-11, MdC2H2-18, MdFKF1-1, and MdFKF1-5 located in the cytoplasm… is good.
A6. We value your suggestion and have restructured the header, table information, and footnotes in Table 1 to better align with the manuscript. Additionally, we have also adjusted the description of these results at line xxx. Reply: Modifications have been implemented in the original text.
Q7. Fig.1
Does tree for C2H2 zinc fingers in accordance with classification from Plant-TFClass, where C2H2 ZF class = C2H2 & IDD families ?
A7. This is the C2H2 ZF class.
Q8. 186
5’ or 3’ UTR ?
A8. This was in fact a typographical error, we have revised the results at line 200.
Q9. 200
on 12 chromosomes -> on twelve chromosomes
(12th chromosome is not twelve chromosomes)
A9. We value your suggestion and have made modifications at line 214.
Q10. Fig.3 201 -215 = too simple and obvious text
It is not biology. It is too descriptive. What is meaning of the exact position of a gene in genome? I think it's almost irrelevant to the biological meaning.
A10. We appreciate your suggestion, these results are merely their physical locations on the genome, irrelevant to the biological issues we are discussing, we have condensed the description at lines 221-224.
Q11. 226
…This suggests that apples have a more advanced evolutionary relationship with the C2H2, DEALL, and FKF1 gene families of Arabidopsis.
What is novelty of this result? E.g. I know that human and mouse are taxonomically closer than human and cat. Try to briefly discuss your result.
A11. We have removed this self-evident statement to enhance the academic nature of the manuscript.
Q12. 232
…start sites of the apple MdC2H2, MdDEALL, and MdFKF1 gene families
Transcription start sites of genes from the families...
A12. These comments pertain to Figure 5, the results related to this figure, specifically all results about the promoter region CRE, have been eliminated in the revised manuscript, the reason is seen in A13.
Q13. Fig. 5 is completely wrong
Axis X must refer to certain transcription factors, in particular, recognition models for their binding sites
You should predict transcription factor binding sites (TFBS) motifs instead of certain ‘boxes’. Use FIMO https://meme-suite.org/meme/tools/fimo, apply TFBS motifs from JASPAR & CISBP, or Plant Cistrome
Rauluseviciute, I., Riudavets-Puig, R., Blanc-Mathieu, R., Castro-Mondragon, J. A., Ferenc, K., Kumar, V., Lemma, R. B., Lucas, J., Chèneby, J., Baranasic, D., et al., (2024) JASPAR 2024: 20th anniversary of the open-access database of transcription factor binding profiles. Nucleic Acids Res, 52(D1), D174-D182
Weirauch, M. T., Yang, A., Albu, M., Cote, A. G., Montenegro-Montero, A., Drewe, P., Najafabadi, H. S., Lambert, S. A., Mann, I., Cook, K., et al. (2014) Determination and inference of eukaryotic transcription factor sequence specificity. Cell 158, 1431–1443.
O'Malley, R.C., Huang, S.C., Song, L., Lewsey, M.G., Bartlett, A., Nery, J.R., Galli, M., Gallavotti, A., Ecker, J.R. (2016) Cistrome and epicistrome features shape the regulatory DNA landscape. Cell 165, 1280–1292.
I suspect that your boxes are something like degenerate consensus or too short matrices, they may have too high false positive rate, you may take all genes for Arabidopsis or apple and test whether any motif is specific for certain gene set compared to gene set of all genes. PlanCARE was published more than 20 years ago, its motifs are not based on modern NGS techniques like ChIP-seq and DAP-seq
Magali Lescot et al. PlantCARE, a database of plant cis-acting regulatory elements and a portal to tools for in silico analysis of promoter sequences Nucleic Acids Res. 2002 Jan 1;30(1):325-327.
A13. Esteemed reviewer, our original intention was to identify the CRE in the promoter region of the identified gene family members, to explore the scenarios in which these genes might be triggered or regulated in apples. However, as you pointed out, PlantCARE is somewhat outdated, hence these results appear meaningless and can easily lead to overinterpretation. The prediction methods you suggested, whether it's the MEME suite component or public databases, they can predict or align the terminal sequences of TFs and target genes, however, these results are merely predictive, highly false positive, and not sufficiently adaptable to plant characteristics. Our results actually include verification of interactions based on molecular experiments, which further diminishes the importance of this part (original Figure 5), therefore we considered deleting all results of this part in the manuscript, we believe such modification is appropriate and highlights the theme. If you believe we still need to make corresponding predictions or other results, we will complete them without reservation, looking forward to your further suggestions!
Once again, we express our gratitude for your insights.
All of these are indispensable for enhancing the quality and publication of our manuscript.
We sincerely hope that these modifications will be acceptable to you. We look forward to your response!
Reviewer 2 Report
Comments and Suggestions for Authors
The authors use advanced molecular techniques and on the other hand make "student" mistakes that disqualify the article. In any scientific study, it is important to correctly state what the research was conducted on so that other researchers can replicate and check it. Unfortunately, from the article by Chuang Mei et al. we do not find out what the authors studied in this way. Please state which species of the genus Malus was analyzed. Was it a pure botanical species or a hybrid?
The second important issue in scientific research is the type and number of samples that are analyzed. Unfortunately, we will not know from the work of Chuang Mei et al. whether the material came from one individual or several. If the material came from one individual then the research should be repeated on at least material from three plants.
Please provide the exact number of buds (how many per developmental stage) were analyzed.
Mallus representatives have already been tested for genes that regulate flower development:
https://www.sciencedirect.com/science/article/pii/S0304416524000369
https://www.ncbi.nlm.nih.gov/pmc/articles/PMC7894833/
https://bmcplantbiol.biomedcentral.com/articles/10.1186/s12870-019-1695-0
Please discuss exactly what has already been discovered and what concretely new this work brings.
Minor comments:
Please write species and generic names using italics.
Author Response
Dear Editors and Reviewer
Thank you for your valuable comments on our manuscript titled “Identification of apple flower development-related gene families and analysis of transcriptional regulation” (Manuscript ID: ijms-2976358). Firstly, we apologize for all the novice errors or formatting issues in our manuscript, we have revised the entire text to ensure its academic quality. Your suggestions have been very helpful in revising and improving our paper and have significant guidance for our research. We have carefully studied these comments and proofread and revised the entire text. The corresponding modifications are marked in red in the revised manuscript, and we hope these changes will be approved by you.
Q1. The authors use advanced molecular techniques and on the other hand make "student" mistakes that disqualify the article. In any scientific study, it is important to correctly state what the research was conducted on so that other researchers can replicate and check it. Unfortunately, from the article by Chuang Mei et al. we do not find out what the authors studied in this way. Please state which species of the genus Malus was analyzed. Was it a pure botanical species or a hybrid?
Q2. The second important issue in scientific research is the type and number of samples that are analyzed. Unfortunately, we will not know from the work of Chuang Mei et al. whether the material came from one individual or several. If the material came from one individual then the research should be repeated on at least material from three plants.
Q3. Please provide the exact number of buds (how many per developmental stage) were analyzed.
A1-3. We appreciate your suggestion and apologize for our rudimentary and information-deficient expression. We have revised the Materials and Methods section at lines 346-354, the information should now be sufficiently clear and reproducible.
Q4. Mallus representatives have already been tested for genes that regulate flower development:
https://www.sciencedirect.com/science/article/pii/S0304416524000369
https://www.ncbi.nlm.nih.gov/pmc/articles/PMC7894833/
https://bmcplantbiol.biomedcentral.com/articles/10.1186/s12870-019-1695-0
Please discuss exactly what has already been discovered and what concretely new this work brings.
A4. We have conducted a comprehensive study of previous research and our results, the discussion section has been thoroughly improved, the corresponding revisions are at lines 485-495.
Q5. Minor comments:
Please write species and generic names using italics.
A5. We have proofread the entire text and revised the italics for Latin and genes.
Once again, we express our gratitude for your insights.
All of these are indispensable for enhancing the quality and publication of our manuscript.
We sincerely hope that these modifications will be acceptable to you. We look forward to your response!

Reviewer 3 Report
Comments and Suggestions for Authors
The authors performed identification of C2H2, DELA, and FKF1 through genome-wide analysis of phylogenetic relationships, physicochemical characteristics, structural characteristics, chromosomal distribution, gene replication and collinearity, cis-acting elements, and gene expression profiles. The manuscript written comprehensively and the theme of the study is presented appropriately. However, I have some suggestions to improve the presentation of this manuscript.
- In introduction, please describe the experimental materials in more detail, why choose ‘apple'’? Also, please explain which genes and proteins have been systematic and comprehensive analysis in apple.
- In results, Fig. 2, 3, 4, and 6 must be high resolution.
- Legend of figure 1, 6, 8, 9 need to be improved.
- In table 1, add chromosome location and explain the full names of No, Mw, AI, and Pii in physicochemical properties.
- In Fig 1, please provide the Gene ID and formal code of the gene.
- In Fig 6, there is a missing Y-axis. And in S1 and S2, there are three bars, what do they mean respectively? Shouldn't they be expressed as mean values?
- In Fig 6A, draw C2H2 as one graph and arrange it in order of gene number.
- In Fig 6, what are the PCR primer sequences?
- In Fig 6, statistical analysis of qPCR results is required.
- In line 399: Discussion instead of conclusion.
. Also, references should be written according to the ‘ijms’ journal format.
Author Response
Dear Editors and Reviewer,
Thank you for your valuable comments on our manuscript titled “Identification of apple flower development-related gene families and analysis of transcriptional regulation” (Manuscript ID: ijms-2976358). Firstly, we apologize for all the novice errors or formatting issues in our manuscript, we have revised the entire text to ensure its academic quality. Your suggestions have been very helpful in revising and improving our paper and have significant guidance for our research. We have carefully studied these comments and proofread and revised the entire text. The corresponding modifications are marked in red in the revised manuscript, and we hope these changes will be approved by you.
Q1. In introduction, please describe the experimental materials in more detail, why choose ‘apple'’? Also, please explain which genes and proteins have been systematic and comprehensive analysis in apple.
A1. Esteemed reviewer, the corresponding explanation has been added to the introduction at lines 36-41.
Q2. In results, Fig. 2, 3, 4, and 6 must be high resolution.
A2. We apologize for the clarity of the images, they have now been remade, and high-resolution images (300dpi or PDF) have been submitted to the review system.
Q3. Legend of figure 1, 6, 8, 9 need to be improved.
A3. We appreciate your opinion, the corresponding changes have been marked in the revised manuscript.
Q4. In table 1, add chromosome location and explain the full names of No, Mw, AI, and Pii in physicochemical properties.
A4. Figure 1 has been upgraded, the corresponding information has been added.
Q5. In Fig 1, please provide the Gene ID and formal code of the gene.
A5. All gene IDs and sequences have been submitted to the review system in the form of supplementary file Annex 1 for your review.
Q6. In Fig 6A, draw C2H2 as one graph and arrange it in order of gene number.
A6. We value your opinion, we have improved the results.
Q7. In Fig 6, what are the PCR primer sequences?
A7. Esteemed reviewer, all primer sequences involved in the manuscript have been uploaded to the review system, see supplementary file Schedule 1.
Q8. In Fig 6, statistical analysis of qPCR results is required. Dpi
A8. We have added statistical analysis of qPCR results.
Q9. In line 399: Discussion instead of conclusion
A9. Sorry for that, we have made revisions.
Q10. Also, references should be written according to the ‘ijms’ journal format.
A10. We have proofread all references and their formats, the citations should now be correct and comply with the 'ijms' journal specifications.
Once again, we express our gratitude for your insights.
All of these are indispensable for enhancing the quality and publication of our manuscript.
We sincerely hope that these modifications will be acceptable to you. We look forward to your response!

Round 2
Reviewer 1 Report
Comments and Suggestions for Authors
About the terminology:
C2H2 family -> C2H2 ZFP family
Jiang, Y., Liu, L., Pan, Z. et al. Genome-wide analysis of the C2H2 zinc finger protein gene family and its response to salt stress in ginseng, Panax ginseng Meyer. Sci Rep 12, 10165 (2022). https://doi.org/10.1038/s41598-022-14357-w
Liu Q, Wang Z, Xu X, Zhang H, Li C (2015) Genome-Wide Analysis of C2H2 Zinc-Finger Family Transcription Factors and Their Responses to Abiotic Stresses in Poplar (Populus trichocarpa). PLoS ONE 10(8): e0134753. https://doi.org/10.1371/journal.pone.0134753
http://www.edgar-wingender.de/Class%202.3.html
Lambert, S.A., Jolma, A., Campitelli, L.F., Das, P.K., Yin, Y., Albu, M., Chen, X., Taipale, J., Hughes, T.R., Weirauch, M.T. (2018) The Human transcription factors. Cell 172, 650–665. https://doi.org/10.1016/j.cell.2018.01.029
English is quite bad, so the sense of many sentences is too ambiguous. Here the Introduction section.
Line 34 misprint
such as difficul y in flowering of young trees
74
transcription factor -> transcription factors
...transcription factor (C2H2, DELLA)... they are families but not factors (I wrote that already)
94
In summary, C2H2 protein is able to promote -> In summary, C2H2 ZF proteins are able to promote
98 DELLA means gene or family ? (I wrote that already) Do you means regulators here?
DELLA (aspartate-glutamate-leucine-leucine-alanine) is a plant-specific transcriptional regulator that mediates gibberellin (GA) signaling.
Currently, the DELLA gene family includes three genes in tobacco, five in Arabidopsis
107
The first DELLA member -> The first member of the DELLA family
164
In summary, C2H2, DELLA, and FKF1 genes are
166
to identify the apple C2H2, DELLA, and FKF1 gene families -> to identify genes from C2H2, DELLA, and FKF1 gene families in apple (here italics Latin name)
I think that these family are already defined for plants. The family is not species-specific term. They common for large taxonomic clades like metazoa or plants
Results, below I don’t care mainly on the quality of English. Since I am not a spell-checker for authors, please invite anyone to correct English
178
The apple genome was comprehensively searched for the C2H2, DELLA, and FKF1 genes using HMM search.
There are no C2H2 or DELLA genes in genome, there are genes from these families there.
It should be as.. We used ……. tools to detect genes from …… families…
Table 1 is still too large. I can allow it to be published. It is the supplementary data. (I wrote that already)
273
conserved motifs of the exon (CDS) and intron (UTR).
I can catch why only exon CDS (triplets), but why only UTR introns?
What does it means? All genes of nuclear genome by definition are in the nuclei. It is a weird section. Do you mean proteins or what?
2.7. Subcellular localization of apple C2H2, DELLA, and FKF1 genes
Where are references in the subsections 3.2, 3.3 and 3.4 ?
641
from the available research -> from the available researches
651
Phylogenetic tree analyses indicate > Phylogenetic tree analyses indicates
678
intron-less -> intronless
We identified 24 C2H2 genes, 6 DELLA genes, and 6 FKF1 genes ->
We identified 24 / 6 / 6 genes from C2H2 / DELLA / FKF1 families
Comments on the Quality of English Language
English requires the most careful check. Many sentences should be completely overwritten.
Author Response
关于术语:
问题1.C2H2 家族 -> C2H2 ZFP 家族
江, Y., Liu, L., Pan, Z. et al.人参C2H2锌指蛋白基因家族及其对盐胁迫的响应的全基因组分析,人参Meyer。科学代表 12, 10165 (2022)。https://doi.org/10.1038/s41598-022-14357-w
Liu Q, Wang Z, Xu X, Zhang H, Li C (2015) 杨树C2H2锌指家族转录因子的全基因组分析及其对非生物胁迫的响应.公共科学图书馆一号10(8):e0134753。https://doi.org/10.1371/journal.pone.0134753
http://www.edgar-wingender.de/Class%202.3.html
Lambert, S.A., Jolma, A., Campitelli, L.F., Das, P.K., Yin, Y., Albu, M., Chen, X., Taipale, J., Hughes, T.R., Weirauch, M.T. (2018) 人类转录因子。单元格172,650-665。https://doi.org/10.1016/j.cell.2018.01.029
答1.我们修改了第48行的相关表述。
问题2.英语相当不好,所以很多句子的意思太模糊了。这里是介绍部分。
答2.我们改进了完整的手稿。
问题3.第 34 行印刷错误
例如幼树开花的困难
答3.我们修改了第39行的相关表述。
问题4.74
转录因子->转录因子
答4.我们修改了第56行的相关表述。
问题5....转录因子(C2H2,DELLA)...他们是家庭,但不是因素(我已经写过了)
答5.我们修改了第56行的相关表述。
问题6.94
总之,C2H2 蛋白能够促进 -> 总之,C2H2 ZF 蛋白能够促进
答6.我们修改了第81行的相关表述。
Q7.98 DELLA的意思是基因还是家族?(我已经写过了)你指的是这里的监管机构吗?
DELLA(天冬氨酸-谷氨酸-亮氨酸-亮氨酸-丙氨酸)是一种介导赤霉素 (GA) 信号传导的植物特异性转录调节因子。
目前,DELLA基因家族在烟草中包括三个基因,在拟南芥中包括五个基因
答7.我们修改了第86行的相关表述。
问题8.107
第一位DELLA成员 -> DELLA家族的第一位成员
答8.我们在第 95 行进行了修订。
问题9.164
总之,C2H2、DELLA 和 FKF1 基因是
答9.我们修改了第127行的相关表述。
问题10.166
鉴定苹果 C2H2、DELLA 和 FKF1 基因家族 ->鉴定苹果中 C2H2、DELLA 和 FKF1 基因家族的基因(此处为斜体拉丁名称)
答10.我们修改了第129行的相关表述。
问题11.我认为这些家族已经为植物定义了。该科不是特定于物种的术语。它们常见于大型分类分支,如后生动物或植物
结果,下面我主要不关心英语的质量。由于我不是作者的拼写检查器,请邀请任何人纠正英语。
答11.我们已尽最大努力修改了英语,但是,我们不是以英语为母语的人,因此我们请求MDPI提供的专业语言修饰的帮助,手稿的写作得到了极大的提高。这是我们所做的修饰的证明
问题12.178
采用HMM搜索法全面检索苹果基因组中的C2H2、DELLA和FKF1基因。
基因组中没有 C2H2 或 DELLA 基因,那里有来自这些家族的基因。
它应该是..我们用过.......从......中检测基因的工具家族。。。
答12.我们修改了第142行的相关表述
问题13.表 1 仍然太大。我可以允许它被发布。它是补充数据。(我已经写过了)
答13.谢谢。我们会带你去的。
问题14.273
外显子 (CDS) 和内含子 (UTR) 的保守基序。
我可以抓住为什么只有外显子 CDS(三联体),但为什么只有 UTR 内含子?
答14.我们修改了第214行的相关表述。
问题15.这是什么意思?根据定义,核基因组的所有基因都在细胞核中。这是一个奇怪的部分。你是说蛋白质还是什么?
2.7. 苹果 C2H2、DELLA 和 FKF1 基因的亚细胞定位
答15.感谢您的内容,此错误的发生是由于在写作过程中手超过头脑。这个结果实际上是指所有蛋白质的亚细胞定位,所以它是蛋白质。在修订后的手稿中,我们对其进行了更正。
问题16.第 3.2、3.3 和 3.4 小节中的参考文献在哪里?
答16.我们修改了第 401,419,425 行中的相关表述。
问题17.641
来自现有研究 - > 来自现有研究
651
系统发育树分析表明>系统发育树分析表明
答17.手稿在第 463 行和第 474 行进行了修订。
问题18.678
无内含子 -> 无内含子
答18.我们修改了第 513 行中的相关表述。
问题19.我们鉴定了 24 个 C2H2 基因、6 个 DELLA 基因和 6 个 FKF1 基因 >
我们从C2H2 / DELLA / FKF1家族中鉴定出24 / 6 / 6个基因
答19.我们修改了第 558 行中的相关表述。
对英语语言质量的评论
英语需要最仔细的检查。许多句子应该被完全覆盖。
Reviewer 2 Report
Comments and Suggestions for Authors
The authors have made some changes, but they are still insufficient to proceed with the article.
The article still contains errors in the spelling of species and generic names in terms of the lack of use of italics.
Please look through all the literature carefully
We do not write the names of the authors of the species and the names of the varieties using italics!
The authors write: "Three flower buds of 10-year old apple trees with uniform size were collected during the dormancy period and the expansion period of flower buds, and 6-8 flower buds were taken from each plant in each period" However, there is still no information on how many trees the material came from.
The authors say they have done a cover review of the literature in terms of apple tree research and modified the discussions. But this is not evident in the paper. How many new literature items were added? I can only see one.
Author Response
Review2
The authors have made some changes, but they are still insufficient to proceed with the article.
Q1. The article still contains errors in the spelling of species and generic names in terms of the lack of use of italics.
Please look through all the literature carefully
We do not write the names of the authors of the species and the names of the varieties using italics!
A1. The incorrect spelling of species and genus names that we corrected.
Q2. The authors write: "Three flower buds of 10-year old apple trees with uniform size were collected during the dormancy period and the expansion period of flower buds, and 6-8 flower buds were taken from each plant in each period" However, there is still no information on how many trees the material came from.
A2. We have amended the relevant expression in line375-379
Q3. The authors say they have done a cover review of the literature in terms of apple tree research and modified the discussions. But this is not evident in the paper. How many new literature items were added? I can only see one.
A3. We have amended the relevant expression in line479-483, 502-504 and 513 – 518.
In the previous revision we deleted some of the content, the references were changed significantly, but no trace of the revision was retained, and the following new references were added:
- [45] Wen, H.; Zhong, W.J.; Huo, X.M.; Zhuang, W.B.; Ni, Z.J.; Gao, Z. H. Expression analysis of ABA- and GA-related genes during four stages of bud dormancy in Japanese apricot (Prunus mume Sieb. et Zucc). J HORTIC SCI BIOTECH. 2016, 91(4), 362-369.
- [46] Gagne, J.M.; Downes, B.P.; Shiu, S.-H.; Durski, A.M.; Vierstra, R.D. The F-box subunit of the SCF E3 complex is encoded by a diverse superfamily of genes in Arabidopsis. Proc. Natl. Acad. Sci. USA 2002, 99, 11519–11524.
- [47] 耆那教,M.;Nijhawan,A.;阿罗拉,R.;阿加瓦尔,P.;雷,S.;夏尔马,P.;卡普尔,S.;Tyagi,又名;Khurana, J.P. 大米中的F-box蛋白。全基因组分析、分类、穗和种子发育过程中的时空基因表达,以及光和非生物胁迫的调控。植物生理学 2007, 143, 1467–1483.
- [48] 贾樟柯;吴,B.;李,H.;黄J.;玉米F-box家族的全基因组鉴定和表征.摩尔热内特。基因。2013, 288, 559–577.

Reviewer 3 Report
Comments and Suggestions for Authors
The authors have clarified most of the questions I raised in my previous review. Now I feel the manuscript could be accepted for publication.
Author Response
感谢您对我们的稿件改进的热情工作!
Round 3
Reviewer 1 Report
Comments and Suggestions for Authors
I read only introduction and it too bad for the third stage of review.
The term 'transcription factor' is not used. This is important since C2H2 ZF & DELLA are families of transcription factors, while the third family FKF1 is not. I asked this in previous comments
I third time insist in English corrections, otherwise I recommend to reject the manuscript.
76
At present, several C2H2, DELLA and 95 FKF1 families regulating flowering have been identified
It is nonsense. There is only one C2H2 ZFP family at least.
77
C2H2 is a class of zinc finger protein with a "finger-like" structure, and its QALGGH 97 domain is an important region that can bind to DNA.
So C2H2 is a class or family.
127 English ?
after the deletion mutant gai-3, especially the anther development of gai-3 leads to male sterility.
132
flowering time under LD in the leaf vascular system,
what is LD ?
140
studies have been shown -> studies have shown
141 SD ?
398
genes from C2H2, DELLA, and FKF1 gene families -> genes from C2H2, DELLA, and FKF1 families
400 Why word properties is used twice here?
properties of the members of the apple C2H2, DELLA, and FKF1 gene families members in phylogeny, chromosomal distribution, physicochemical properties
401
delete 'and motif prediction', you did not perform it.
411
The apple C2H2, DELLA, and FKF1 genes -> The genes from C2H2, DELLA, and FKF1 families.
I cannot read this manuscript anymore, I cannot be a corrector in almost any line. There are still too many misprints. Authors do not possess nor English nor, formal logic of usual language.
Comments on the Quality of English LanguageIt seems that authors are too far from even formally correct English. English as well as logic of sentences is too bad
Author Response
尊敬的审稿人:
感谢您对我们题为“Identification of Apple Flower Development-Related Gene Families and Analysis of Transcriptional Regulation”的稿件提出的宝贵意见。你的所有建议都是明确、直接和建设性的。遗憾的是,我们在学术写作和英语方面的弱点导致我们的稿件没有达到出版的最低质量要求,尽管进行了两次重大修改。
在仔细研究了您的评论后,我们根据之前的修改完全提升了手稿的所有部分。这包括英语写作错误、观点描述不明确、专业词汇使用不当以及引言和讨论的结构。
我们担心的是,手稿被修改得如此之多,以至于由于修改标记过多而不再可读。因此,在新修订版中,我们完全删除了修订标记(尽管它是基于前两个修订版修改的)。我们不确定 MDPI 或审稿人是否会容忍此操作。
总之,我们已经进行了彻底的更改。我们诚恳地恳请审稿人再给我们一次机会,再次评估我们的稿件。我们非常希望这些变化在学术上是充分的,并且你能接受。
真诚地
所有作者
Reviewer 2 Report
Comments and Suggestions for Authors
The authors have made a number of changes and improved the article. But there are still many errors in the literature in terms of writing generic and species names. Please Remember that Arabidopsis is a generic name and should be written in italics.
"Borkh. cv. Red Fuji" please write without italics!
Author Response

(The authors gave the same response as above.)
